# Understanding Trauma in IPV: Distinguishing Complex PTSD, PTSD, and BPD in Victims and Offenders

**DOI:** 10.3390/brainsci14090856

**Published:** 2024-08-25

**Authors:** Erica Pugliese, Federica Visco-Comandini, Carolina Papa, Luciana Ciringione, Lucia Cornacchia, Fabiana Gino, Loreta Cannito, Stefania Fadda, Francesco Mancini

**Affiliations:** 1Associazione di Psicologia Cognitiva APC e Scuola di Psicoterapia Cognitiva SPC, 00185 Roma, Italy; federica.visco@gmail.com (F.V.-C.); carolina.papa@uniroma1.it (C.P.);; 2Department of Psychology, Sapienza University of Rome, 00185 Roma, Italy; 3Department of Psychology and Cognitive Science, University of Trento, 38068 Rovereto, Italy; 4Scuola di Psicoterapia Cognitiva, 37122 Verona, Italy; 5Società Italiana di Cognitivismo Clinico, 00185 Roma, Italy; 6Associazione Scuola di Psicoterapia Cognitiva, 58100 Grosseto, Italy; 7Department of Social Sciences, University of Foggia, 71122 Foggia, Italy; 8Department of Human Sciences, Guglielmo Marconi University, 00193 Roma, Italy

**Keywords:** cPTSD, PTSD, BPD, intimate partner violence, IPV victims, IPV offenders, review

## Abstract

This work aims to shed light on the differential diagnosis of complex post-traumatic stress disorder (cPTSD), post-traumatic stress disorder (PTSD), and borderline personality disorder (BPD) within the context of intimate partner violence (IPV), which represents a highly innovative field of clinical research. To this end, a critical review of the literature was conducted to identify and compare the clinical patterns and symptomatic overlaps among cPTSD, PTSD, and BPD, with an emphasis on their manifestation in both IPV victims and offenders. The results show that despite some symptomatic similarities, cPTSD, PTSD, and BPD have distinct clinical patterns of interpersonal violence. Specifically, disturbances in self-organization (DSO) are more commonly found in offenders, while the diagnosis of cPTSD seems more aligned with the psychological functioning of victims. In addition, cPTSD and specific characteristics of BPD, such as fear of rejection and instability of identity, constitute risk factors for IPV victimization. cPTSD is shown as a predisposing factor not only for IPV victims but also for offenders, while PTSD emerges as a consequential factor. The specific pathways linking PTSD, cPTSD, and BPD with IPV have significant implications for clinical practice. Further research is needed to understand these profiles and the mechanisms linking trauma-related features to IPV, which is crucial for implementing effective violence prevention programs.

## 1. Introduction

The phenomenon of intimate partner violence (IPV) has gathered significant attention due to its harmful effects on victims and society. According to the WHO, 30% of women globally aged 15 to 70 have experienced some form of physical and/or sexual violence by an intimate male partner in their lifetime [1]. The prevalence of this phenomenon varies by region, with 33% of women in Africa, the Eastern Mediterranean, and Southeast Asia, 25% in the Americas, 22% in Europe and high-income countries, and 20% in the Western Pacific [1]. 

It is essential to acknowledge that these statistics exclusively consider women as victims. However, both women and men can be victims of IPV, as there can be both male and female offenders. For example, in a study conducted in Ireland [2], 32.1% of the participants reported experiencing lifetime IPV, with a higher prevalence among females (ORs ranging from 1.51 to 2.18). Specifically, IPV impacted approximately one in three females and one in four males in Ireland. Latent class analysis results revealed that the risk factors for females included younger age, having children (AOR = 4.28), lower income level (AOR = 0.80), reduced social support (AOR = 0.94), and limited social contact (AOR = 0.85). On the other hand, the risk factors for males were living in an urban environment (AOR = 3.01), having children (AOR = 4.13), and lower social support (AOR = 0.96) [2]. The study found that all instances of IPV exposure significantly increased the likelihood of experiencing multiple suicide-related phenomena for both males and females [2]. In support of this, a study investigating offending behaviors found no main effect for gender [3]. Furthermore, gender differences have been found to vary depending on the types of offending behaviors and maltreatment profiles. For example, a study [4] found that the risk of juvenile delinquency increased significantly for women who had foster care experiences with frequent placement changes. An analysis of data from the National Violent Death Reporting System in the United States found that 20% of suicides were related to individuals who were experiencing intimate partner problems such as breakups, conflict, divorce, and IPV [5]. Those who have undergone IPV have an increased risk of suffering from a range of mental health problems, such as post-traumatic stress disorder, substance abuse, depression, anxiety, suicidal thoughts, and behaviors, but subsequent suicide attempts have only been found among women and not in male IPV survivors [2]. Circumstances associated with an increased likelihood of suicide related to intimate partner problems include interpersonal violence, victimization, financial problems, occupational problems, and family problems. In contrast, suicides not associated with intimate partner problems are more likely to occur in older people and to be caused by health problems or crime. In addition, researchers have found that the link between suicide and relationship problems may be bidirectional: a violent and dysfunctional couple relationship may exacerbate mental health problems that may contribute to difficulties in an intimate relationship with a partner [5].

In the last few years, new research has shed further light on IPV cultural and social dimensions. Emerging perspectives emphasize the significance of cultural factors in understanding IPV dynamics. According to Green and colleagues [6] cultural norms and values profoundly impact the manifestation and reporting of IPV, necessitating culturally sensitive approaches in research and practice. Moreover, a current research on sexual objectification in romantic relationships reveals that such objectification, directly and indirectly, shapes attitudes toward dating violence, highlighting that its effects extend beyond individuals to influence broader relationship dynamics [7]. Although some research has identified the condition of unemployment and poverty [8,9], internalized social norms [10], and the presence of substance use disorders [11,12] as risk factors for violent behavior regardless of gender [13], over time, there has arisen an ever-increasing need to understand the psychological vulnerability factors associated with violence in intimate relationships [14]. Studies on the psychological aspects have mainly delved into how childhood trauma influences adult IPV victimization and perpetration, leading to suggestions for targeted intervention strategies. Recent meta-analyses [15,16] have provided more profound insights into these dynamics, emphasizing the interplay between early trauma and later IPV experiences suggesting targeted intervention strategies. Furthermore, one of the psychological variables that has been identified as a risk factor for violence in intimate relationships is pathological affective dependence (PAD) [17], a relational condition in which one or both partners adopt violent, controlling, abusive, or manipulative behavior toward the other and the relationship generates suffering in at least one of the two partners. According to the PAD theory [17], this condition emerges from the dissatisfaction of some basic needs in early caring relationships. Specifically, the dissatisfaction of the need for love, dignity, and safety with caregivers combined with the dysfunctional beliefs ingrained in our society leads people with PAD to seek this satisfaction in problematic and abusive relationships actively but, despite this, fail to leave their partners [18]. Regardless of the relationship between such adverse early experiences and PAD, the study of the processes and mechanisms involved in fostering the development of this risky psychological condition for IPV is still very much lacking.

The PAD condition fuels the perpetuation of the cycle of violence in intimate relationships, opening a fundamental question in IPV research relating to what extent early relational factors contribute to favoring and maintaining interpersonal violence over time. According to the PAD theory, experiencing or perpetrating violence in a relationship could be a coping mechanism to manage the unpleasant emotions arising from the relationship with caregivers, which are then replicated in the relationship with the partner. Various research indicates that the victim and the abuser share the same vulnerability factors, deriving from adverse early experiences of a relational nature [19,20]. That insecure attachment is associated with a greater risk of IPV victimization, revictimization, and perpetration [21,22]. These relationships negatively affect mental and physical health [23,24,25,26,27].

Recently, evidence has revealed that IPV victims reported increased symptoms of post-traumatic disorder [28]. Although the research in recent years has been interested in differentiating between post-traumatic stress disorder (PTSD), complex post-traumatic stress disorder (cPTSD), and borderline personality disorder (BPD) to distinguish the specific characteristics of these functioning profiles—which have a significant impact on individual well-being [29,30]—no study has attempted to make this distinction by focusing on how these characteristics differently impact violence in intimate relationships.

To address this gap, this review aims to critically examine and compare the clinical patterns and symptomatic overlaps between cPTSD, PTSD, and BPD, focusing on their manifestation in both victims and offenders of IPV. We will first present the distinctions between cPTSD, PTSD, and BPD, as documented in the literature, and then discuss how these differences specifically manifest in the context of IPV.

## 2. Distinctive Features of Complex PTSD, PTSD, and BPD

Since the introduction of the diagnosis of PTSD, it has been recognized that trauma takes on more severe and intricate forms when endured over prolonged periods [31]. PTSD is characterized by disruptions in traumatic memory processing, resulting in intrusive thoughts, flashbacks, and nightmares [32]. Additionally, dysregulated neurotransmitter systems, particularly serotonin and norepinephrine, contribute to heightened arousal and emotional numbing in PTSD patients [33].

Nowadays, the diagnosis of PTSD in the *DSM-5* [34] falls short of capturing and elucidating the complex symptomatology stemming from chronic traumatic experiences [35]. In that sense, Herman proposed the first conceptualization of cPTSD [36], highlighting the interpersonal nature of severe and repeated traumatic experiences. This new definition emphasizes two key aspects: firstly, the societal framework enabling the exploitation of a marginalized group, and secondly, the relational aspect of the trauma. Indeed, it illuminates a condition of captivity, subjected to the control and dominance of a perpetrator. According to the *11th International Classification of Diseases* (ICD-11), cPTSD is characterized by a constellation of symptoms that extends beyond those of PTSD. These include affect dysregulation, negative self-concept, and disturbances in relationships, which often arise from prolonged or repetitive trauma, such as childhood abuse or prolonged domestic violence [37]. The *ICD-11* emphasizes that cPTSD involves profound changes in self-perception and interpersonal functioning, often resulting from experiences where escape is not possible. This was an essential innovation in the field of traumatic stress research. Several prolonged controversies surround the actual characteristics of the nosological status and composition of the proposed cPTSD construct [38,39,40,41]. Its validity as a clinical syndrome has been questioned, primarily due to overlapping symptomatology with other trauma-related disorders [42].

Based on this idea, a comprehensive definition of complex traumatization has been proposed, emphasizing the nature of the complex trauma within developmental stressors. According to this perspective, traumatic stressors are characterized by being (1) repetitive and prolonged, (2) involving direct and indirect harm and/or neglect and abandonment by caregivers, (3) occurring during developmental phases of vulnerability, and (4) posing a significant threat to a child’s development trajectory [43]. However, evidence suggests that while developmental trauma increases the risk of developing PTSD, it is not necessarily a prerequisite [44].

Both PTSD and cPTSD diagnoses are now categorized under the general classification of “disorders specifically associated with stress” [45,46]. This classification helps to classify the trauma-related symptoms without implying a static and unmodifiable disposition, as with the term “personality disorder”, but rather as a set of symptoms that may lead to change.

To delineate the specific nature of cPTSD, we will outline the main differences between (1) cPTSD and PTSD and (2) cPTSD and BPD. Afterward, we will compare these profiles concerning violence in intimate relationships and concerning IPV victims and offenders. Indeed, those diagnoses are often described in the literature as having overlapping clusters and symptoms [47] with consequences on the chosen treatment.

### 2.1. Complex Post-Traumatic Stress Disorder and Post-Traumatic Stress Disorder

The diagnostic criteria for cPTSD have undergone evolution over time, with the latest description in the *ICD-11* [37] and a considerable amount of research and clinical evidence that points out the need for a differentiated diagnosis between PTSD and cPTSD.

A DSM-5 team specializing in PTSD has identified 27 main symptoms and proposed an expanded diagnostic category, which includes complex PTSD, emphasizing disturbances in self-organization (DSO) such as affective dysregulation, negative self-concept, and relational disturbances [34]. Even the WHO’s *ICD-11* now includes a distinction between the diagnosis of PTSD and cPTSD, which encompasses the three clusters of diagnostic criteria for PTSD (i.e., re-experience of trauma, avoidance of trauma-related stimuli, and a sense of current threat). Additionally, the presence of other psychopathological elements complicates prognosis and treatment, further impairing the individual’s functionality in different areas (e.g., work and relationships). These symptoms are defined as DSO, including affective dysregulation, negative self-concept, and relationship disturbances [29].

Psychological elements that could characterize cPTSD [35,36] include the following: (1) Exposure to severe trauma and chronic, prolonged, and repeated interpersonal abuse; (2) attachment failure, which is typical in the life story of people with cPTSD, along with several episodes of repeated traumatization in childhood; (3) inadequate sense of self, altered patterns, emotional dysregulation, and impulse control [48]; (4) poorer treatment adherence and outcomes, with challenges in achieving effectiveness (5) worse prognosis and an extended course, also due to heightened functional impairment; (6) major comorbidities, (such as anxiety disorders, mood disorders, somatization disorders, and personality disorders) [48], than individuals who did not meet the criteria for PTSD; and (7) higher risk factors for psychopathology (e.g., dissociation, self-injurious behaviors, and substance abuse). These characteristics led researchers to define eight symptom clusters that characterize cPTSD [49]:Affective dysregulation (e.g., shifts in affective regulation that may occur as enduring feelings of dissatisfaction, tendencies towards self-harm or suicidal thoughts, explosive or notably restrained anger, compulsive or inhibited sexual behaviors, and/or suppressed or unpredictable emotional responses);Behavioral dysregulation (e.g., difficulties in controlling impulses, violence towards others, and/or risky behaviors);Impairments in interpersonal relationships (i.e., avoidance, isolation and withdrawal, disruption in intimate relationships, repeated search for a helper with pervasive or dysfunctional demands for care and reassurance, persistent distrust, and/or repeated failures of self-protection);Attentional or monitoring difficulties in the ability to direct or shift attention away from trauma-associated stimuli;Dissociation—alterations in consciousness (e.g., amnesia or hypermnesia due to traumatic events, transient dissociative episodes, and/or depersonalization/derealization);Somatic suffering (e.g., chronic pain, difficulty in regulating nervous system activation);Dissociative identity symptoms (i.e., altered self-concept with extremely fluctuating, unstable, and chaotic representations;Altered meaning systems (i.e., negative self-concept symptoms are defined as persistent beliefs about oneself as belittled, defeated, or worthless and are accompanied by deep and pervasive feelings of shame, guilt, or failure) [45,46]. Affective dysregulation, negative or altered self-concept, and disturbances in relationships are the three additional clusters of symptoms that, according to *ICD-11*, reflect disorders in self-organization [45,50,51].

Various studies have investigated the vulnerability factors predisposing to the development of cPTSD. Among these, early experiences of torture, interpersonal violence, neglect, abuse, genocide [52], traumatic bereavement, domestic or IPV [14,17,18,53], institutional abuse, e.g., that which may occur within foster care [54], or traumatic experiences in war refugees [55] were identified as relevant factors in developing cPTSD [56].

PTSD is a potential clinical outcome after encountering a traumatic stressor, delineated in the *ICD-11* [37] by three primary criteria: reliving the event, such as via flashbacks and nightmares; avoiding reminders; and experiencing a prevailing sense of imminent danger that is often characterized by heightened vigilance. Unlike the *ICD-11*, the latest edition of the *DSM-5-TR* [34] does not include a diagnosis specifically for cPTSD. Indeed, it acknowledges that there is a substantial heterogeneity in the symptoms observed in traumatized individuals, expanding the range and types of symptoms included in the PTSD diagnosis.

For instance, the *DSM-5* introduces a symptom cluster related to negative alterations in mood and cognition, along with a dissociative subtype, to address certain aspects of affect disturbance and self-perception [34,57]. However, the authors stress that these expansions have raised concerns about the practicality of the diagnosis due to the potential for generating numerous symptom profiles under a single diagnosis and the challenges in translating a diagnosis into treatment planning [57]. The findings from Hyland and colleagues [44] indicate that the revised model of psychotraumatology proposed for *ICD-11* establishes a more stringent criterion for diagnosis compared to the *DSM-5*.

While both systems generally agreed on who should receive a diagnosis, there was a notable subset of individuals who met the criteria for PTSD diagnosis according to *DSM-5* but not under *ICD-11*. Specifically, PTSD symptomatology includes the typical eight symptom clusters [30], also shared with other types of disorders (e.g., mood disorders, personality disorders). The overlap primarily focuses on deficits in interpersonal functioning, emotion regulation, and self-perception [30,34,37,58]. The *ICD-11* cPTSD diagnosis includes six symptom clusters, highlighting key distinctions from PTSD.

While three clusters align with PTSD symptoms (re-experiencing, avoidance, and sense of threat), cPTSD introduces three additional clusters related to DSO, specifically addressing affect dysregulation, negative self-concept, and relationship difficulties. The differentiation between PTSD and cPTSD has garnered support from various researchers. According to Brewin and colleagues [59], several studies have identified at least two distinct symptom profiles. One profile characterizes a group with elevated levels of symptoms across all six clusters of cPTSD (re-experiencing, avoidance, sense of threat, affect dysregulation, negative self-concept, and disturbances in relationships). In contrast, another profile reflects high levels of PTSD symptoms but low levels of symptoms related to disturbances in DSO.

Recent studies investigating the discriminant validity of cPTSD in refugees found a two-class solution through latent class analysis, supporting different psychopathological profiles among PTSD and cPTSD [55,59,60,61]. These results add to the growing empirical literature supporting discrimination between PTSD and cPTSD in samples from culturally and trauma-diverse backgrounds [61]. These findings suggest that there are apparent differences in the characteristics of each disorder.

### 2.2. Borderline Personality Disorder and Complex Post-Traumatic Stress Disorder

The accuracy and utility of clinical assessments for adults who experienced chronic childhood maltreatment are often compromised by clinicians’ inability, due to a lack of clarity, to address complex psychological functioning, frequently resulting in comorbid diagnoses [62,63,64]. Incorrect formulations can hinder the delivery of safe and effective treatments [65,66]. These adults commonly receive multiple comorbid diagnoses, especially with BPD and PTSD [64,67]. According to the *DSM-5*, borderline personality disorder (BPD) is defined as “a pervasive pattern of instability of interpersonal relationships, self-image, and affects, and marked impulsivity, beginning by early adulthood and present in a variety of contexts” [34]. The etiological factors for BPD are multifaceted, involving genetic predisposition, neurobiological abnormalities, and early life adversity, which contribute to its complex clinical presentation [68].

A recent systematic review by Atkinson and colleagues [69] showed that most studies found different cPTSD and BPD profiles. Only one study involving a population without major trauma displayed no differences between the two constructs [70]. Indeed, the study by Owczare et al. [71], which used network analysis to examine the relationships between ICD-11-PTSD symptoms, DSO, and BPD in a clinical sample of polytraumatized individuals, showed that BPD and cPTSD are primarily distinct and that their symptom groups overlap only minimally. The only overlap between the two was found in the symptoms of “Affective Dysregulation”, symptoms linked to BPD. This study adds to the evidence for the discriminant validity of cPTSD and emphasizes its uniqueness from BPD. Finally, the review by Stopyra et al. [72] aimed to qualitatively compare neuroimaging findings on effect, attention, and memory processing in cPTSD, PTSD, and BPD and showed that these disorders might represent a spectrum in which similar brain regions are involved. However, differences in activation patterns could explain their unique symptom manifestations. The authors concluded that neural changes in these disorders can be better understood by examining a symptom-based continuum underlying cPTSD, PTSD, and BPD.

Indeed, looking at the two profiles (cPTSD and BPD), emotional dysregulation occurs in both cPTSD and BPD, but while there is a chronic difficulty in finding comfort when distressed in cPTSD, in BPD, there is extreme and uncontrolled anger and profound emotional dyscontrol [73].

As for anger, suicidal and self-injurious behaviors occasionally occur in cPTSD, while they are more central and frequent in BPD [29]. In cPTSD, the negative perceptions of self-experience tend to center around a chronic sense of guilt, shame, and worthlessness [74], in contrast to a more unstable and fragmented sense of self-present in BPD. While both BPD and cPTSD entail severe relationship challenges, they manifest differently in terms of relational patterns. In BPD, there is a pronounced reactive hostility within relationships, often accompanied by a cycle of intense attachment and detachment to avoid perceived abandonment [29]. Individuals with BPD often have an overwhelming need for closeness and may exhibit demanding behaviors to fulfill this need, while in cPTSD, dysregulation is characterized by both avoidance and detachment, rooted in the fear of proximity and intimacy with others [59,75]. The fear of intimacy may moderate the need for closeness, leading individuals to cope by maintaining distance in relationships, perceiving them as too risky. This fear of intimacy is systematically accompanied by a continuous perception of experienced betrayal and a severe emotional detachment within relationships [50] and secondary feelings of sadness due to the failure to achieve affective and interpersonal goals. They may desire a relationship, but shame and worry about burdening others prevent them from pursuing one. In contrast, the desire for closeness in BPD takes the form of intense anger and restraint, coupled with a terror of abandonment. They often oscillate between demanding closeness and resorting to impulsive threats of abandonment to avoid being left.

As Ford and Courtois summarized, “hypervigilance related to being harmed” would be at the core of cPTSD, while “extreme sensitivity (which can take the form of hypervigilance) to perceiving oneself as abandoned” would be at the heart of the BPD [47]. In addition to analyzing symptom overlap across the constructs of cPTSD, PTSD, and BPD, it is also essential to identify their common characteristics. Impairments in interpersonal relationships and social emotions (e.g., feelings of guilt, shame, and self-blame) are components of these three disorders [47]. Some evidence in the PTSD population confirmed that there is a prevalence of shame, self-blame, and guilt [76,77], as well as the ways interpersonal dysfunction exacerbates PTSD by increasing social isolation [78].

High levels of shame and self-blame also mark BPD, and individuals with cPTSD are described as experiencing pervasive difficulties in relationship functioning [17,18,43]. These findings are in line with the notion that trauma exposure or neglected childhood environments alter the interpersonal system, with negative consequences on emotions related to self-perception, as well as on the ability to relate to and trust in others [36,75,79]. Therefore, these symptoms and those of cPTSD are connected with the exact life domains (i.e., affect regulation, relationships, and self-beliefs) typically affected [80].

Saraiya and colleagues [70] found that individuals with PTSD, cPTSD, and BPD have the highest levels of psychological distress, traumatic event history, adverse childhood experiences, and PTSD symptoms. However, shame, a central emotion in trauma that can differentiate its severity, was the only social emotion to differ between them [70,74] significantly.

In addition, dissociation has been associated with cPTSD [81] and BPD likewise [82]. These similarities have prompted some authors to suggest reclassifying BPD as a trauma-related disorder [83]. Although BPD and cPTSD have some similarities (as do cPTSD and PTSD), it is not appropriate to consider cPTSD as a subtype of BPD. Evidence suggests that a sub-group of BPD patients, who often but not always have comorbid PTSD, may be best understood and treated if cPTSD is explicitly addressed alongside BPD [47]. A better differentiated empirically grounded view of cPTSD, BPD, and PTSD is a high priority for the advancement of clinical practice and research with traumatized adults.

In a recent review, Paris [84] found some difficulties in reconceptualizing some cases of BPD within the newer diagnosis of cPTSD. The cPTSD construct focuses on the role of childhood trauma in shaping relational problems in adulthood. However, according to the author, this concept does not consider the role of gene–environment interactions that would instead support a biosocial theory of BPD [84]. The cPTSD model fails to include the role of heritable personality traits as an element of psychosocial risk factors.

In the following section, we will delineate the differences between cPTSD, PTSD, and BPD within the framework of IPV. The focus is on the differences delineated in the literature between offenders and victims concerning various subcomponents of traumatic symptoms, types of violence, victim/offender roles, and symptom classes.

## 3. IPV as a Cross-Cutting Factor between PTSD, cPTSD, and BPD

Considering the distinct symptom clusters of cPTSD, PTSD, and BPD, we explore how these profiles are differently associated with IPV and how traumatic characteristics differ in victims and perpetrators (see Table 1).

It is well known that early adverse and traumatic experiences lead to difficulties in intimate and interpersonal relationships in general [85], representing a significant risk factor for anxiety and depressive disorders in adulthood [86,87]. Regarding the possible mechanisms that mediate this relationship, several studies have focused on the role of impairment in emotional regulation resulting from parents’ derogatory and denigrating behaviors towards the child and the consequent negative emotions repeated over time [88,89]. Emotional regulation difficulties affect interpersonal functioning and are negatively associated with warmth, assertiveness, positive relationships, and intimacy [90,91].

Concerning the association between cPTSD and IPV, Karatzias and colleagues [92] stated that cPTSD appears to be significantly associated with maladaptive regulation strategies. Children raised in turbulent, unpredictable, or unsupportive environments develop specific strategies for managing their emotions to adapt to the environment (e.g., avoidance). While these strategies may be adaptive in the short term, they interfere with long-term adaptation in the broader relational context [93]. Furthermore, in response to interpersonal trauma, negative beliefs about oneself can combine with the negative evaluation of others (e.g., that they are dangerous or unreliable), contributing to feelings of threat and paranoia [94]. The perception of others as dangerous can increase the risk of violence to manage the perceived threat [95]. Indeed, the results of the MacArthur study on violence risk revealed that suspiciousness significantly predicts subsequent violent behavior, including physical and verbal aggression [96]. In line with this, recent research has shown how self-hatred, a dimension significantly present in trauma [97], mediates the relationship between paranoia and hetero-directed hostility, and this relationship increases depending on how much the individual feels deserves self-persecution [98].

A mechanism underlying cPTSD that has been associated with IPV consists of attachment disorganization and role reversal experiences during childhood that would lead individuals to be unable to assert their needs in interpersonal relationships, increasing the risk of victimization [99]. Furthermore, it is not merely the experience of traumatic interpersonal events themselves but rather the lack of integration of these events that appears crucial in predisposing individuals to IPV. This is because abuse and maltreatment in childhood may have been denied, preserving psychological integrity at that time, but leading to similar dysfunctional tendencies in adult relationships [100].

Moreover, these findings are equally factual for individuals with PTSD, who exhibit a higher likelihood of perceiving unrealistic threats and a more hostile evaluation of events [101]. Some studies on veterans have suggested that such impairments explain the dysregulation of anger and the perpetration of both physical and psychological IPV in PTSD [102]. Research on PTSD and IPV has focused more on the development of traumatic symptoms following partner victimization and as a risk factor for revictimization. Alterations at the psychological, biological, neurological, physiological, and behavioral levels have been found in victims because of IPV [103]. Therefore, a recent study has investigated how PTSD symptoms promoted revictimization, finding that disengaged coping led to a much higher risk of partner revictimization at a six-month follow-up [104]. This finding is consistent with the inability to separate from the partner present in victims of violence [18]. The dimension of PTSD that has shown the strongest associations with IPV is hyperarousal, which predisposes to violence towards the partner through various pathways (e.g., sleep problems related to hyperarousal) [105]. Recent research, however, has also found moderate associations with emotional numbing [106], which could be an essential risk factor for IPV due to the depletion of internal resources due to the effort to avoid emotions associated with trauma.

Several studies have investigated the psychological consequences of IPV to understand whether it is differentially linked to PTSD rather than cPTSD symptomatology. One study comparing the presence of PTSD and cPTSD symptoms in women victims of violence found that the prevalence of cPTSD was twice that of PTSD, with high levels of fear associated with re-experiencing, avoidance, sense of current threats and disturbances in relationships [107]. In contrast, a study investigating the prevalence of traumatic symptoms among male perpetrators of IPV in Israel found a high prevalence of traumatic events and PTSD symptoms [31]. Interestingly, the authors found that cumulative lifetime trauma was associated with PTSD symptoms, while cumulative childhood violence was associated with the DSO cluster in perpetrators of IPV.

Another study also found that in men who were perpetrators of violence, the component of DSO was preeminently compared to PTSD symptoms, which were still present as an effect of childhood victimization [108]. Indeed, DSO problems are associated with severe issues in intimate partner relationships, including psychological violence as both victim and perpetrator and the perpetration of sexual IPV [109]. This result indicates that this dimension is the most connected to psychological violence in the partner’s intimate relationships in the form of emotional abuse and alternating relational instability and disengagement [110]. It is possible, therefore, that the perception of one’s internal states and those of the partner as uncontrollable or dangerous leads to the implementation of solution attempts based on control and power in the relationship, leading to higher levels of violence.

Another study conducted by Dyer and colleagues [111] found that physical aggression was the most frequent form of violence, significantly associated with cPTSD concerning PTSD. Furthermore, high hostility is present in both PTSD and cPTSD, reflecting attitudes of bitterness and resentment [112]. Hostility may take on different meanings in these two profiles: in PTSD, it could be a defensive response associated with the idea of the other as unpredictable, while in cPTSD, there could be the addition of a component linked to the desire to obtain compensation because of the trauma [14,18].

Regarding the relationship between BPD and IPV, this diagnosis seems to be present in both men and women who perpetrate violence [12,113] rather than in victims. It shows strong associations with different types of IPV (i.e., psychological, physical, and sexual) [114]. Research by Munro and Selbom [115] further investigated how BPD traits, when considered at a dimensional level, were associated with various forms of IPV. They found that hostility was more linked with the physical and psychological forms of IPV, while risk-taking and suspiciousness were related to the physical and sexual forms.

It is essential to underline that BPD has also been found in victims of IPV but in the form of personality traits and, in any case, associated with PTSD [116]. Furthermore, the characteristics of BPD that constitute a more significant risk factor for victimization are fear of rejection, loneliness, and identity instability [117]. Pugliese [17] showed that under the condition of IPV, victims behave as if they have a personality disorder, as the offender’s behavior is a trigger of their dysfunctional traits. These dysfunctional traits disappear when they are out of the violent condition. Even a study that evaluated the predisposition to IPV in borderline personality functioning did not find a mediating role in sensitivity to rejection but rather in anger, which is a characteristic equally present in cPTSD [118]. As with the other profiles, in BPD, there are specific functioning characteristics that are associated differently with IPV and with victims vs. offenders. Indeed, affective instability and interpersonal disorders (e.g., separation concerns) play an important role in IPV perpetration, while identity disorders play an essential role in IPV victimization [117]. These data are in line with the results that emerged about PAD, a significant risk factor for IPV, characterized by an unstable self-image [18].

In general, BPD has been studied more in offenders than in victims, so it is possible that the data available to date are not fully explanatory of its relationship with IPV. These results highlight how IPV emerges as a phenomenon that is linked to various psychological factors, taking on specific characteristics about the type of violence and the role of the offender rather than the victim. Furthermore, the research highlights how IPV can be expressed differently in PTSD, cPTSD, and BPD profiles, starting from vulnerability factors that include individual traumatic experiences and the specific sequelae connected to them and are accompanied by equally peculiar cognitive-affective patterns and behaviors that favor violence in intimate relationships.

## 4. Discussion

Evidence on cPTSD still faces challenges in distinguishing this disorder from others that appear similar, such as PTSD and BPD [64]. The overlap between these three diagnostic categories and their symptoms and the resulting inaccuracy in clinical assessments compromises the efficacy of treatments [65,66,119].

This critical review had two main objectives: (1) to contribute to the current literature by addressing the ongoing scientific debate surrounding these three main psychological conditions, which are often confused or considered interchangeable, and (2) to investigate if and how these siblings’ diagnoses are differently associated with the complex condition of IPV.

Accordingly, this article initially describes the specific clinical characteristics of the cPTSD, focusing on the recent literature. The results show that cPTSD is characterized by affective dysregulation, behavioral dysregulation, impairments in interpersonal relationships, attentional difficulties to stimuli related to the trauma, dissociation, somatic distress, dissociative identity symptoms, and altered self-perception. Then, the focus is to understand the differences and similarities between cPTSD and the two related disorders (PTSD and BPD), which manifest overlapped clusters of symptoms. The results show that differently from PTSD, the *ICD-11* cPTSD diagnosis includes six symptom clusters: three overlaps with PTSD (re-experiencing, avoidance, and sense of threat) and three additional clusters related to DSO (affect dysregulation, negative self-concept, and relationship difficulties).

Regarding the relationship between cPTSD and BPD, the results indicate some overlapping symptoms, such as emotional dysregulation, hyperarousal, and difficulties in interpersonal relationships. While these diagnoses intersect in some areas, several essential differentiating elements exist. Precisely, patients’ coping mechanisms to deal with these symptoms are different.

For instance, emotional dysregulation in cPTSD individuals is expressed in a chronic difficulty in finding comfort when distressed, while BPD expresses extreme and uncontrolled anger as an external mechanism [73]. The emotions experienced during emotional dysregulation in cPTSD are linked to the sense of guilt, shame, and worthlessness, while BPD shows a more fragmented and disrupted sense of self [74]. As for intense hyperarousal, cPTSD is related to avoidance following an intrusive reliving of traumatic memories and is related to a persistent fear of tomorrow, with also intense emotional suffering and a self-perception of uselessness, shame, and guilt. These symptoms are systematically accompanied by a continuous fear of intimacy, a perception of experienced betrayal, and a severe emotional detachment within relationships [50] that results in an intense emotion of sadness.

In BPD, the hyperarousal passes through the perception of anger in combination with a fear of abandonment and impulsive acting out in relationships. Moreover, in cPTSD, the difficulties at the interpersonal levels seem to be characterized by avoidance and detachment based on fear of the proximity and intimacy of the other [59,75]. At the same time, BPD patients show reactive relational hostility, alternating entanglement, and disengagement to avoid real or imagined abandonment [29].

Finally, the main aim of this work was to investigate how the sibling diagnoses of cPTSD, PTSD, and BPD are related within the context of IPV and whether they are associated differently with the roles of the IPV victim or offender. This critical review shows that, even concerning the relationship of these profiles with IPV, some elements in common between the three diagnostic categories can be found, while others are explicitly characterizing for each category.

The component of suspiciousness that predisposes to IPV is present in the three diagnoses but in different forms. In cPTSD, it is connected to the fear of intimacy and closeness in the form of a fear of being able to relive the traumatic experience [50], while in PTSD, it is more linearly associated with impairments in the interpretation of events [101]. On the contrary, in BPD, suspiciousness falls within the dimensional conceptualization of the disorder [115]. However, it is probably more connected to the fear of rejection, which would risk confirming a belief of being unworthy.

Another IPV-related characteristic that the profiles have in common is anger. However, while PTSD and cPTSD typically manifest as internalized interpersonal hostility, in BPD, it is more uncontrolled and externally expressed [73]. It is no coincidence that this type of externalized hostility in BPD is more connected to offenders than to victims of IPV [120] and has frequently been found in women who perpetrate violence [12,113]. In contrast, men who perpetrate violence appear to be a more significant contributor to the DSO component compared to the other factors [108], with DSO emerging as the strongest predictor of severe forms of IPV, including sexual IPV [109]. Regarding forms of violence, it has been found that physical violence is more connected to cPTSD [111]. At the same time, psychological and sexual abuse are more connected to DSO [109,110], while all forms of IPV are associated with BPD [114].

A risk factor for IPV, however, appears to be the instability of the sense of self, which is associated with a greater probability of victimization [117] and is a factor also present in PAD, a psychological condition found in victims of violence [17]. Regarding PTSD, the specific characteristic that seems to predispose to IPV concerns hyperarousal following the trauma [105], and this data seems consistent with the idea that the activation resulting from a perceived sense of threat can more easily lead to defense responses that can take on violent characteristics. PTSD symptoms have been associated more with IPV victims than perpetrators [104], although they have also been found in some studies involving IPV perpetrators [31,108]. However, it must be considered that in these studies, PTSD was investigated as opposed to cPTSD in offenders [31] and that the DSO component was the most associated with forms of violence related to abuse [109].

The most relevant distinction that emerges in the declination of trauma in IPV is as follows. With some exceptions, in the context of IPV, cPTSD acts as a predisposing factor for both victims and offenders. In contrast, PTSD is identified more as a psychopathological consequence of IPV and as a risk factor for re-traumatization. BPD, on the other hand, overlaps with cPTSD in constituting a critical risk factor for IPV but is more about the violence perpetrated than the violence suffered. The dimensions of BPD seem to be across the board for victims and offenders. However, the systematic review by Guzmán and colleagues [120] reveals mixed results with gender differences. BPD was found more in offenders, especially female offenders, but there is also a disparity in the number of studies that considered BPD in victimization. Furthermore, it is essential to note that BPD is a disorder that is more prevalent in women than in men, accounting for 75% of individuals with this diagnosis [34]. BPD characteristics consisting of alcohol abuse and hostility are associated with a risk of perpetration [118,121], whereas both fear of rejection and identity instability for victimization are seen in IPV [117].

Considering these results, it is possible to conclude that the difference in IPV is associated with the amount of BPD symptoms and the consequent functional impairments. Additionally, many studies investigating the associations with IPV have not accounted for the differentiation between different profiles of traumatic events and how the overlap of symptoms and clusters is expressed in IPV. In BPD, hostility is most strongly associated with the physical and psychological forms of IPV, whereas risk-taking and suspiciousness are associated with physical and sexual forms [115]. This result is consistent with the known relationship between suspicions of infidelity and impulsive responses underlying gender-based violence perpetrated by men [122]. However, no study has investigated the same relationship in women.

Moreover, DSO represents a hybrid psychological cluster that can be considered as both a predisposing and consequent factor for IPV. This can be explained by considering the transdiagnostic role of DSO symptoms among the three psychopathologies. In conclusion, the overlap between PTSD, cPTSD, and BPD is an issue of clinical relevance that poses several treatment-related questions that research is beginning to answer. The peculiar characteristics of these profiles begin to emerge about the various factors that concern vulnerability, cognitive-affective states, coping, and the consequent psychological suffering. One of the most relevant aspects is the link that each of these profiles has with IPV, a phenomenon that has a massive impact on health globally [1]. The main studies considered in the review related to the relationship between trauma and IPV are given in Appendix A (Table A1). 

## 5. Conclusions

The data confirm how IPV constitutes a multifactorial phenomenon that can take on phenomenologically diverse characteristics and how much these characteristics are influenced by specific psychological functioning that emerges in response to trauma. Considering these distinctions and outlining increasingly comprehensive paths of the peculiar cognitive and behavioral mechanisms involved in IPV is fundamental if we want to counteract a phenomenon of this magnitude on both a psychological and social level. The fact that PTSD, cPTSD, and BPD are associated differently with IPV should, therefore, open a new research question regarding which of these conditions is associated with more significant harmful outcomes.

Several gaps remain. For example, even if it is known that suspicions of infidelity are among the main factors that predispose to violence and homicide in intimate relationships [122], there are no studies that have investigated in which cases jealousy can take pathological forms or be associated with specific psychological functioning that increase the risk of IPV. According to Dutton and colleagues [123], anger, jealousy, BPD organization, and trauma symptoms are significantly correlated with the frequency of perpetration of verbal and physical IPV. However, there is very little research on this topic. Similarly, some cognitive processes or dysfunctional relational coping that may promote IPV have also not been investigated in their associations with the PTSD, cPTSD, and BPD conditions, e.g., angry rumination or revenge. Although it provides new and exciting perspectives on the relationship between trauma and IPV, the review is not without limitations; as this is a critical review of the literature, sophisticated methods of collecting the studies considered that would have strengthened the conclusions of this study were not used: this severely limits the replicability and reliability of the study. However, this work has highlighted that no experimental studies have provided information on the incidence rates of these disorders in IPV and have not compared them in victim and offender samples. The lack of clarity on these mechanisms and their relationship with IPV leaves open the possibility that there are further traumatic profiles related to this phenomenon with equally specific features that need to be investigated. A potential fourth condition, PAD, might encompass aspects of the three diagnoses but is specific to IPV [14,17,18]. PAD could indeed explain the underlying psychological reasons for the inability to separate from an abusive partner, a difficulty shared by the victim profiles of all three disorders in the context of IPV. However, PAD has not yet been included in the diagnostic criteria, although the attention of clinicians and professionals has grown. Pugliese and colleagues [18] describe PAD as a relational dynamic where at least one partner suffers due to the abusive behaviors of the other. It is primarily marked by an internal conflict between the desire to separate and the need to save the relationship at all costs, coupled with the perception of an inability to leave an abusive partner. Since PAD is present in victims, it may be equally important to investigate its counterpart, known as counter-dependency, in its relationship with IPV. Indeed, both conditions underline a difficulty in regulating dependency needs in intimate relationships, with possible implications for relational satisfaction and different impacts on IPV. Studies capable of measuring and intervening early in both PAD and counter-dependency are crucial if we genuinely want to contribute to counteract the phenomenon of IPV. Expanding research to include these new diagnostic considerations and investigating their unique contributions to IPV can lead to more effective prevention and intervention strategies, ultimately reducing the prevalence and impact of IPV on individuals and society.

## Figures and Tables

**Table 1 brainsci-14-00856-t001:** Relationship between PTSD, cPTSD, BPD, and IPV.

Disorder	General Characteristics (Core Symptoms and Features of Each Disorder)	Associations with IPV (How the Disorder Influences or Is Influenced by IPV)	Victim Profile (Common Traits or Behaviors of Victims with the Disorder)	Perpetrator Profile (Common Traits or Behaviors of Perpetrators with the Disorder)
PTSD	- Hyperarousal - Emotional numbing - Difficulty disengaging from abusive relationships	- Revictimization due to emotional numbing and hyperarousal - Aggression linked to threat perception	- Hyperarousal - Emotional numbing - Struggles to leave abusive situations	- Aggression driven by heightened vigilance - Perceives constant threats
cPTSD	- Avoidance - Emotional detachment - Disturbance in self-organization (DSO)	- Emotional detachment and avoidance lead to revictimization - Severe relational instability and psychological violence in perpetrators	- Avoidance - Emotional detachment - High revictimization risk	- Relational instability - Psychological violence - Aggression fueled by paranoia and self-hatred
BPD	- Emotional volatility - Identity instability - Fear of abandonment	- High risk of both victimization and perpetration - Strong links to IPV through emotional volatility and impulsivity	- Fear of rejection - Identity instability - High vulnerability to IPV	- Impulsivity - Aggression (psychological, physical, sexual) - Intense, unstable relationships

Note. IPV = intimate personal violence; PTSD: post-traumatic stress disorder; cPTSD: complex post-traumatic stress disorder; BPD: borderline personality disorder; DSO: disturbance in self-organization.

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
