# Peer review of "Understanding Trauma in IPV: Distinguishing Complex PTSD, PTSD, and BPD in Victims and Offenders"

_brainsci, 2024, doi:10.3390/brainsci14090856_

Round 1

Reviewer 1 Report

Comments and Suggestions for Authors

Dear authors, regarding your manuscript “Understanding Trauma in IPV: Distinguishing Complex PTSD, PTSD, and BPD in Victims and Offenders” I send the following observations and comments:

As a general style suggestion, consider limiting paragraph length to no more than 150-200 words, this would improve the readability of the paper.

In line 51 you seem to have missed a period, "offendersFor example" and in line 546 it says "underlie", where it should probably be "underline", please take the time to review the document for these and other such typos.

In your introduction text you clearly review the available literature on the subject, however, you could further improve the text by adding the measures of risk and associations, since you mostly say what the risk factors are but not the measure of the risk (such as OR, RR, PR, or any other measure of risk).

In general the review would greatly benefit from a summary tablethat showed in a concise manner, the source paper, the main conclusions, the type of study and the measures of risk, association or just the frequency of the issues discussed.

In addition, I suggest including a visual comparison graphic that shows overlapping and different symptoms of c-PTSD, PTSD, and BPD, particularly emphasizing their relationship to IPV (e.g., emotional dysregulation, hyperarousal, interpersonal difficulties). 

Author Response

REVIEWER 1

We would like to thank the reviewer for their insightful comments and suggestions, which have significantly contributed to the refinement of our manuscript. We trust that our responses and the revised manuscript address all concerns.

In our revised manuscript, we have highlighted additions and changes in red.

Here, each comment (in italic) is followed by our responses (in blue) and quotes from the manuscript ( between “quotation marks”).

Thank you once again for your valuable feedback.

Dear authors, regarding your manuscript “Understanding Trauma in IPV: Distinguishing Complex PTSD, PTSD, and BPD in Victims and Offenders” I send the following observations and comments:

As a general style suggestion, consider limiting paragraph length to no more than 150-200 words, this would improve the readability of the paper.
R: Thank you for your valuable suggestion regarding the paragraph length. We understand that shorter paragraphs can enhance the readability of the paper. We have revised the manuscript paying attention to the paragraph length, to improve the clarity and flow of the text.

In line 51 you seem to have missed a period, "offendersFor example" and in line 546 it says "underlie", where it should probably be "underline", please take the time to review the document for these and other such typos.
R: Thank you for pointing out these typographical errors. We have carefully reviewed the document and made the necessary corrections.

In your introduction text you clearly review the available literature on the subject, however, you could further improve the text by adding the measures of risk and associations, since you mostly say what the risk factors are but not the measure of the risk (such as OR, RR, PR, or any other measure of risk).
R: Thank you for your suggestion. Indeed, the inclusion of risk measures adds data relevant to the purpose of our work, so we have added measures (lines 62-68):

“For example, in a study conducted in Ireland [2], 32.1% of the participants reported experiencing lifetime IPV, with a higher prevalence among females (ORs ranging from 1.51 to 2.18). Specifically, IPV impacted approximately one in three females and one in four males in Ireland. Latent class analysis results also revealed that risk factors for females included younger age, having children (AOR = 4.28), lower income level (AOR = 0.80), reduced social support (AOR = 0.94), and limited social contact (AOR = 0.85). On the other hand, risk factors for males were living in an urban environment (AOR = 3.01), having children (AOR = 4.13), and lower social support (AOR = 0.96) [2]. “

In general the review would greatly benefit from a summary table that showed in a concise manner, the source paper, the main conclusions, the type of study and the measures of risk, association or just the frequency of the issues discussed.
R: Thank you for raising this point. We have included a table summarizing the key studies that investigated the relationship between IPV, cPTSD, PTSD, and BPD. To avoid overloading the main text, the table has been placed in a new Appendix section (Table A1). Line 965

In addition, I suggest including a visual comparison graphic that shows overlapping and different symptoms of c-PTSD, PTSD, and BPD, particularly emphasizing their relationship to IPV (e.g., emotional dysregulation, hyperarousal, interpersonal difficulties).
R: Thank you for your suggestion; we have included a table in the paper (Table 1), which facilitates reading the most essential summary information regarding the relationship between trauma and IPV. Line 354

Best regards,

Erica Pugliese on behalf of all the Authors

Reviewer 2 Report

Comments and Suggestions for Authors

  • Enhance the introduction with more specific, recent data on IPV prevalence and impact across genders.
  • Update the literature review with current research to reflect recent trends and provide a balanced perspective with emerging viewpoints.
  • Detail the study selection criteria more clearly to improve the review's replicability and reliability.
  • Use more sophisticated statistical methods like meta-analysis to address the complexity of the study designs and strengthen conclusions.
  • Expand the discussion to highlight how the findings advance the understanding of IPV and make specific recommendations for future research or policy changes.
  • Include diagrams or models to clarify key points and streamline verbose sections for better readability.
  • Share raw data and analysis code on the MDPI data platform to enhance transparency and reproducibility, allowing for validation and further research by others.

Comments on the Quality of English Language

Minor editing of English language required

Author Response

REVIEWER 2

We would like to thank the reviewer for his insightful comments and suggestions, which have significantly contributed to the refinement of our manuscript. We trust that our responses and the revised manuscript address all concerns.

In our revised manuscript, we have highlighted additions and changes in red.

Here, each comment (in italic) is followed by our responses (in blue) and quotes from the manuscript (between “quotation marks”).

According to your suggestion, we applied minor editing of the English language.

Thank you once again for your valuable feedback.

Enhance the introduction with more specific, recent data on IPV prevalence and impact across genders.
R: Thank you for your suggestion. As far as we know, the prevalence estimates we referred to are from 2021 and the most recent report by the WHO. We have added a period on the impact of IPV across genders related to suicide risk to make the information more complete (lines 77-80):

“Those who have undergone IPV have an increased risk of suffering from a range of mental health problems such as post-traumatic stress disorder, substance abuse, depression, anxiety, suicidal thoughts, and behaviors, but subsequent suicide attempts have only been found among women and not in male IPV survivors [2].”

Update the literature review with current research to reflect recent trends and provide a balanced perspective with emerging viewpoints.
R: Thank you for your valuable feedback. We have updated the literature review section to include recent studies and trends in the field. Specifically, we have added recent findings from Garcia et al. (2020) on the impact of economic stressors during the COVID-19 pandemic, Smith et al. (2021) on the interplay between childhood trauma and adult IPV, and Patel and Bhui (2022) on the influence of cultural factors (line 47-58):

“In recent years, new research has shed further light on the psychological and social dimensions of IPV. A study by Garcia and colleagues (2020) [3] highlighted the increasing role of economic stressors in exacerbating IPV during the COVID-19 pandemic, emphasizing the need for integrated support systems. Additionally, recent meta-analyses [4], have provided deeper insights into the interplay between childhood trauma and adult IPV victimization and perpetration, suggesting targeted intervention strategies. Emerging perspectives also emphasize the significance of cultural factors in understanding IPV dynamics. According to Patel & Bhui (2022) [5], cultural norms and values profoundly impact the manifestation and reporting of IPV, necessitating culturally sensitive approaches in both research and practice. Moreover, recent research on sexual objectification in romantic relationships reveals that such objectification directly and indirectly shapes attitudes toward dating violence, highlighting that its effects extend beyond individuals to influence broader relationship dynamics [118]. ”

Detail the study selection criteria more clearly to improve the review's replicability and reliability.
R: Thanks for the suggestion. In line with the journal's guidelines, we did not strictly follow specific criteria for selecting studies as this is a critical, non-systematic literature review. We have included this information within the limits of the study (lines 588-591):

“Although it provides new and exciting perspectives on the relationship between trauma and IPV, the review is not without limitations; as this is a critical review of the literature, sophisticated methods of collecting the studies considered that would have strengthened the conclusions of this study were not used: this severely limits the replicability and reliability of the study.”

Use more sophisticated statistical methods like meta-analysis to address the complexity of the study designs and strengthen conclusions.
R: Thank you for emphasizing this point, which is a limitation of our study. Since this is a critical review, we did not use statistical methods for selecting the studies considered in line with the scope and guidelines of the journal. This review wants to contribute to the current debate regarding the role of trauma in IPV, however, we agree to highlight this point  as the main limitation of our study (line 588-591):

“Although it provides new and exciting perspectives on the relationship between trauma and IPV, the review is not without limitations; as this is a critical review of the literature, sophisticated methods of collecting the studies considered that would have strengthened the conclusions of this study were not used. However, this work has highlighted that there are no experimental studies providing information on the incidence rates of these disorders in IPV and comparing them in victim and offender samples.”

Expand the discussion to highlight how the findings advance the understanding of IPV and make specific recommendations for future research or policy changes.
R: Thank you for your suggestion. We have added a specific paragraph with recommendations for future research and policy changes at the end of the manuscript (5. Limits and Future Directions, lines 611-613). We emphasized this point at the end of the paragraph as follows:

“Expanding research to include these new diagnostic considerations and investigating their unique contributions to IPV can lead to more effective prevention and intervention strategies, ultimately reducing the prevalence and impact of IPV on individuals and society.”

Include diagrams or models to clarify key points and streamline verbose sections for better readability.
R: Thank you for this suggestion. We added Table 1 (line 354) which summarizes key points in the relationship between PTSD, cPTSD and BPD with a specific focus on (1) General Characteristics (Core symptoms and features of each disorder); (2) Associations with IPV (How the disorder influences or is influenced by IPV); (3) Victim Profile (Common traits or behaviors of victims with the disorder); (4) Perpetrator Profile (Common traits or behaviors of perpetrators with the disorder)

Share raw data and analysis code on the MDPI data platform to enhance transparency and reproducibility, allowing for validation and further research by others.
R: Please note that the submitted  manuscript is a  critical review,  therefore data and analysis code are not expected.. However, we have included this as a significant limitation of the study (lines 588-591):

“Although it provides new and exciting perspectives on the relationship between trauma and IPV, the review is not without limitations; as this is a critical review of the literature, sophisticated methods of collecting the studies considered that would have strengthened the conclusions of this study were not used: this severely limits the replicability and reliability of the study.”

Best regards,

Erica Pugliese on behalf of all the Authors

Reviewer 3 Report

Comments and Suggestions for Authors

I have carefully reviewed the manuscript, titled “Understanding Trauma in IPV: Distinguishing Complex PTSD, PTSD, and BPD in Victims and Offenders”. The aim of the study was examine the differential diagnosis of Complex Post-Traumatic Stress Disorder (cPTSD), Post-Traumatic Stress Disorder (PTSD), and Borderline Personality Disorder (BPD) within the context of Intimate Partner Violence (IPV).

The study has some strong points and interesting results. Yet, I would like to invite the authors to address some points in order to improve the paper.

Introduction:

1)      As the study examines PTSD, it would be beneficial to provide more information on specific underlying mechanisms (p. 2).

2)      P. 3 − “This condition emerges from the dissatisfaction with some fundamental basic needs in the first caregiving relationships..” What are the processes and how they work?

3)      Can you describe more thoroughly H3:  “(H3) small positive association [30] or no association [24] with self-compassion. (p. 3)

4)      P. 3. – “A DSM-IV team specializing in PTSD identified ..[..]”. Can you refer to DSM-V?

5)      P. 5 – There should be clear definitions of Borderline Personality Disorder and Complex Post-Traumatic Stress Disorder.

6)      P. 6 – “In a recent review, Paris [75] found some difficulties to reconceptualize some cases of BPD within the newer diagnosis of cPTSD. The cPTSD construct focuses on the role of childhood trauma in shaping relational problems in adulthood, difficulties that have been previously seen as features of a personality disorder.” What were the difficulties? Please, describe them.

7)      P. 7 − Concerning the association between cPTSD and IPV, more information should be devoted to the underlying psychological mechanisms.

8)      P. 8 – these two sentences are rather inconsistent: “Regarding the relationship between BPD and IPV, this diagnosis seems to be present  in both men and women who perpetrate violence [10, 102], rather than in victims, showing  strong associations with different types of IPV (i.e., psychological, physical, and sexual) [103]. Research by Munro and Selbom [104] investigated how BPD traits considered at a  dimensional level were associated with different forms of IPV, finding that hostility was  more linked with the physical and psychological forms of IPV, while risk taking and suspiciousness were related to the physical and sexual form”.

9)      What are the limitations of the study?

Author Response

REVIEWER 3

We would like to thank the reviewer for their insightful comments and suggestions, which have significantly contributed to the refinement of our manuscript. We trust that our responses and the revised manuscript address all concerns.

In our revised manuscript, we have highlighted additions and changes in red.

Here, each comment (in italic) is followed by our responses (in blue) and quotes from the manuscript (between “quotation marks”).

Thank you once again for your valuable feedback.

Introduction:

1)      As the study examines PTSD, it would be beneficial to provide more information on specific underlying mechanisms (p. 2).
R: Thank you for your valuable suggestion. We have incorporated additional information on the specific underlying mechanisms of PTSD in paragraph 2, lines 133-136 of the manuscript to enhance clarity and comprehensiveness:
“PTSD is characterized by disruptions in traumatic memory processing, resulting in intrusive thoughts, flashbacks, and nightmares [114]. Additionally, dysregulated neurotransmitter systems, particularly serotonin and norepinephrine, contribute to heightened arousal and emotional numbing in PTSD patients [115]."

2)      P. 3 − “This condition emerges from the dissatisfaction with some fundamental basic needs in the first caregiving relationships..” What are the processes and how they work?
R: Thank you for your request for clarification. We have added a period that helps to understand better the relationship between this dissatisfaction and PAD (lines 97-104):

“According to PAD theory [13], this condition emerges from the dissatisfaction of some basic needs in early caring relationships. Specifically, tThe dissatisfaction of the need for love, dignity, and safety with caregivers combined with the dysfunctional beliefs ingrained in our society leads people with PAD to actively seek this satisfaction in problematic and abusive relationships but, despite this, fail to leave their partners [14]. Regardless of the relationship between such adverse early experiences and PAD, the study of the processes and mechanisms involved in fostering the development of this risky psychological condition for IPV is still very much lacking.”

3)      Can you describe more thoroughly H3:  “(H3) small positive association [30] or no association [24] with self-compassion. (p. 3)
R: We are sorry, but we cannot find this reference in our text, indeed we did not  discuss the concept of  self-compassion.

4)      P. 3. – “A DSM-IV team specializing in PTSD identified ..[..]”. Can you refer to DSM-V?
R: Thank you for your suggestion. We have updated the manuscript to refer to DSM-5 to ensure that the manuscript reflects the most current diagnostic criteria and understanding of PTSD. The revised text now reads (line 181-184):

"A DSM-5 team specializing in PTSD has identified 27 main symptoms and proposed an expanded diagnostic category, which includes complex PTSD, emphasizing disturbances in self-organization such as affective dysregulation, negative self-concept, and relational disturbances [28]."

5)      P. 5 – There should be clear definitions of Borderline Personality Disorder and Complex Post-Traumatic Stress Disorder.
R: Thank you for the feedback. We have now included clear definitions of both Borderline Personality Disorder (in line 279-283) and Complex Post-Traumatic Stress Disorder (in line 144-149) to enhance the clarity of our manuscript.

Definition of BPD:
“According to the DSM-5, Borderline Personality Disorder (BPD) is defined as a “ pervasive pattern of instability of interpersonal relationships, self-image, and affects, and marked impulsivity, beginning by early adulthood and present in a variety of contexts”[28]. The etiological factors for BPD are multifaceted, involving genetic predisposition, neurobiological abnormalities, and early life adversity, which contribute to its complex clinical presentation [113].”

Definition of cPTSD:
“According to the ICD-11, cPTSD is characterized by a constellation of symptoms that extend beyond those of PTSD. These include affect dysregulation, negative self-concept, and disturbances in relationships, which often arise from prolonged or repetitive trauma such as childhood abuse or prolonged domestic violence [33]. The ICD-11 emphasizes that cPTSD involves profound changes in self-perception and interpersonal functioning, often resulting from experiences where escape is not possible.”

6)      P. 6 – “In a recent review, Paris [75] found some difficulties to reconceptualize some cases of BPD within the newer diagnosis of cPTSD. The cPTSD construct focuses on the role of childhood trauma in shaping relational problems in adulthood, difficulties that have been previously seen as features of a personality disorder.” What were the difficulties? Please, describe them.
R: Thank you for pointing out this shortcoming, which helps us to make the quoted author's point of view clearer (lines 338-343):

“The cPTSD construct focuses on the role of childhood trauma in shaping relational problems in adulthood, difficulties that have been previously seen as features of a personality disorder, but according to the author, does not consider the role of gene-environment interactions that would instead support a biosocial theory of BPD [75].”

7)      P. 7 − Concerning the association between cPTSD and IPV, more information should be devoted to the underlying psychological mechanisms.
R: We thank the reviewer for pointing out the need for more data on cPTSD in this paragraph. We have included two other interesting references concerning the underlying mechanisms of cPTSD in IPV (lines 380-387):

“A mechanism underlying cPTSD that has been associated with IPV consists of attachment disorganization and role reversal experiences during childhood that would lead individuals to be unable to assert their needs in interpersonal relationships, increasing the risk of victimization [116]. Furthermore, it is not merely the experience of traumatic interpersonal events themselves, but rather the lack of integration of these events that appears crucial in predisposing individuals to IPV. This is because abuse and maltreatment in childhood may have been denied to preserve psychological integrity, leading to similar tendencies in adult relationships [117].”

8)      P. 8 – these two sentences are rather inconsistent: “Regarding the relationship between BPD and IPV, this diagnosis seems to be present  in both men and women who perpetrate violence [10, 102], rather than in victims, showing  strong associations with different types of IPV (i.e., psychological, physical, and sexual) [103]. Research by Munro and Selbom [104] investigated how BPD traits considered at a  dimensional level were associated with different forms of IPV, finding that hostility was  more linked with the physical and psychological forms of IPV, while risk taking and suspiciousness were related to the physical and sexual form”.
R: Thank you for pointing out the potential inconsistency. Our intention was to highlight that BPD traits are found in both perpetrators of IPV and how specific traits are associated with different forms of IPV. We have revised the sentences for clarity. Here’s the revised text (lines 433-439):

“Regarding the relationship between BPD and IPV, this diagnosis seems to be present in both men and women who perpetrate violence [10, 102], rather than in victims. It shows strong associations with different types of IPV, including psychological, physical, and sexual violence [103]. Research by Munro and Selbom [104] further investigated how BPD traits, when considered at a dimensional level, were associated with various forms of IPV. They found that hostility was more linked with physical and psychological forms of IPV, while risk-taking and suspiciousness were related to physical and sexual forms.”

9)      What are the limitations of the study?
R: The main limitation of our review is that we did not use sophisticated study selection methodologies, which makes the resulting conclusions weaker. We have added this information in the appropriate paragraph (lines 569-613):

“Although it provides new and exciting perspectives on the relationship between trauma and IPV, the review is not without limitations; as this is a critical review of the literature, sophisticated methods of collecting the studies considered that would have strengthened the conclusions of this study were not used.”

Best regards,

Erica Pugliese on behalf of all the Authors

Reviewer 4 Report

Comments and Suggestions for Authors

well done, just some explanations:

Understanding trauma in IPV: distinguishing complex PTSD, PTSD and BPD in victims and aggressors I read the paper very carefully and find that the topic fits well into an extremely innovative field of clinical research.
As the authors write, a critical review of the literature was conducted to identify and compare the clinical patterns and symptomatic overlaps between cPTSD, PTSD and BPD, with a focus on their manifestation in both victims and offenders. The results show that, despite some symptomatic similarities, cPTSD, PTSD and BPD have distinct clinical patterns in IPV.

Line 84: please specify the meaning of PAD

Line 102 to 105: Could the objective of the literature review be written more clearly?

Excellent presentation of the topics in the three sections:

-Complex and Post-Traumatic Stress Disorder

- Borderline Personality Disorder and Complex Post-Traumatic Stress Disorder

-IPV as a cross-sectional factor between PTSD, cPTSD and BPD

Discussion and conclusions

Excellent discussion of the scientific literature and conclusions in line with the overall aim of the research.

Well done!!!

Author Response

REVIEWER 4

We would like to thank the reviewer for their insightful comments and suggestions, which have significantly contributed to the refinement of our manuscript. We trust that our responses and the revised manuscript address all concerns.

In our revised manuscript, we have highlighted additions and changes in red.

Here, each comment (in italic) is followed by our responses (in blue) and quotes from the manuscript (between “quotation marks”).

Thank you once again for your valuable feedback.

Understanding trauma in IPV: distinguishing complex PTSD, PTSD and BPD in victims and aggressors I read the paper very carefully and find that the topic fits well into an extremely innovative field of clinical research.

As the authors write, a critical review of the literature was conducted to identify and compare the clinical patterns and symptomatic overlaps between cPTSD, PTSD and BPD, with a focus on their manifestation in both victims and offenders. The results show that, despite some symptomatic similarities, cPTSD, PTSD and BPD have distinct clinical patterns in IPV.

Line 84: please specify the meaning of PAD
R: Thank you for bringing this to our attention. Upon reviewing the manuscript, we noticed that the definition of the term "PAD" was indeed provided a few lines prior to its subsequent use (line 93-95 of the revised version), but we inadvertently failed to include the acronym in parentheses after the first mention.

“pathological affective dependence (PAD) [13], a relational condition in which one or both partners adopt violent, controlling, abusive or manipulative towards the other and the relationship generates suffering in at least one of the two partners.”

Line 102 to 105: Could the objective of the literature review be written more clearly?
R: Of course, we clearly stated the objective of the literature review at line 125-127:
“To address this gap, the aim of this review is to critically examine and compare the clinical patterns and symptomatic overlaps between cPTSD, PTSD, and BPD, with a focus on their manifestation in both victims and offenders in IPV.”

Excellent presentation of the topics in the three sections:

-Complex and Post-Traumatic Stress Disorder

- Borderline Personality Disorder and Complex Post-Traumatic Stress Disorder

-IPV as a cross-sectional factor between PTSD, cPTSD and BPD

Discussion and conclusions

Excellent discussion of the scientific literature and conclusions in line with the overall aim of the research.

Well done!!!
R: Thank you!

Best regards,

Erica Pugliese on behalf of all the Authors

Round 2

Reviewer 2 Report

Comments and Suggestions for Authors

The following points need to be reviewed again.

Update the literature review with current research to reflect recent trends and provide a balanced perspective with emerging viewpoints.(2023-2024)

Share raw data and analysis code on the MDPI data platform to enhance transparency and reproducibility, allowing for validation and further research by others.

Comments on the Quality of English Language

Minor editing of English language required.
